# Tobacco and Pituri Use in Pregnancy: A Protocol for Measuring Maternal and Perinatal Exposure and Outcomes in Central Australian Aboriginal Women

**DOI:** 10.3390/mps2020047

**Published:** 2019-06-07

**Authors:** Angela Ratsch, Kathryn Steadman, BoMi Ryu, Fiona Bogossian

**Affiliations:** 1Research Education, Development and Support, Wide Bay Hospital and Health Service, Hervey Bay 4655, Australia; 2School of Pharmacy, The University of Queensland, Brisbane 4072, Australia; k.steadman@uq.edu.au (K.S.); ryu.bomi@gmail.com (B.R.); 3School of Health & Sports Science and School of Nursing, Midwifery and Paramedicine, University of the Sunshine Coast, Maroochydore 4558, Australia; fbogossi@usc.edu.au; 4School of Nursing, Midwifery and Social Work, The University of Queensland, Brisbane 4072, Australia

**Keywords:** smokeless tobacco, chewed tobacco, antenatal, pregnancy, pregnancy outcomes, tobacco and nicotine concentrations, Indigenous Australian, Aboriginal

## Abstract

Maternal tobacco smoking is a recognized risk behavior that has adverse impacts on maternal and fetal health. However, in some populations, the use of smokeless tobacco exceeds the use of smoked tobacco. In central Australia, Aboriginal populations utilize wild tobacco plants (*Nicotiana* spp.) as a smokeless product. These plants are known by a variety of names, one of which is pituri. The plants are masticated and retained in the oral cavity for extended periods of time and their use continues throughout pregnancy, birth, and lactation. In contrast to the evidence related to combusted tobacco use, there is no evidence as to the effects of pituri use in pregnancy. Central Australian Aboriginal women who were at least 28 weeks pregnant were stratified into three tobacco exposure groups: (a) Pituri chewers, (b) smokers, and (c) non-tobacco users. Routine antenatal and birth information, pre-existing and pregnancy-related maternal characteristics, fetal characteristics, and biological samples were collected and compared. The biological samples were analysed for tobacco and nicotine metabolite concentrations. Samples from the mother included venous blood, urine, hair and colostrum and/or breast milk. From the neonate, this included Day 1 and Day 3 urine and meconium, and from the placenta, arterial and venous cord blood following delivery. This is the first study to correlate the pregnancy outcomes of central Australian Aboriginal women with different tobacco exposures. The findings will provide the foundation for epidemiological data collection in related studies. **Note to readers:** In this article, the term “Aboriginal” was chosen by central Australian women to refer to both themselves and the Aboriginal people in their communities. “Indigenous” was chosen to refer to the wider Australian Aboriginal and Torres Strait Islander people.

## 1. Introduction

“Tobacco” is a common name for the genus *Nicotiana* plant and its products utilized in cigars, cigarettes, pipe tobacco, snuff, and chewing tobacco (CT) [1]. In Australia, 22 native species and four sub-species of *Nicotiana* are widely dispersed across the continent (Figure 1 [2]).

The major pharmacologically active alkaloid of *Nicotiana* spp. is nicotine [3], a dose-dependent potent and lethal poison. Nornicotine, anatabine, and anabasine are minor alkaloids present in most tobacco species [3] and similarly, have dose-dependent lethal effects [4]. In addition to these compounds, tobacco-specific nitrosamines are formed from tobacco and nicotine products and are responsible for the formation of carcinogenic tobacco-specific nitrosamines including N’-nitrosonornicotine and 4-(methylnitrosamino)-1-(3-pyridyl)-1-butanone [5]. In this research, tobacco-specific nitrosamines are not examined.

Worldwide, tobacco is most widely known and utilized as a personal nicotine-delivery product. For this purpose, it is generally either burnt, including as cigarettes, cigars, pipes, bidis and in hookahs, or used as nasal and/or oral smokeless tobacco (ST) including as snuff, snus, tooth powder, twist, loose leaf, dissolvable and CT [6]. Nicotine is also the principal pharmacologically active component in a range of heat-not-burn tobacco products, pharmaceutical nicotine replacement therapy products [7] and electronic cigarette devices [8].

### 1.1. Tobacco and Nicotine Pharmacodynamics and Pharmacokinetics

#### 1.1.1. Nicotine Maternal, Placental, Fetal, and Breast Milk Distribution

Following maternal exposure to tobacco and/or nicotine, nicotine readily crosses the placental barrier and is concentrated in fetal and placental compartments [9]. The ratio of fetal/maternal serum ng/mL from smoked tobacco demonstrates a higher concentration of nicotine in the fetal compartments than in the maternal serum (placental tissue 2.58 ± 1.30:1, amniotic fluid 1.54 ± 0.27:1 and umbilical vein 1.12 ± 0.30:1 [10]). Significantly, the fetal nicotine concentration is not dependent upon the time since the last maternal nicotine exposure or sampling interval [10]. Continued nicotine administration to the neonate post-birth is possible through colostrum and/or breast milk. The breast milk pH varies between 6.75–7.42 [11] and nicotine is rapidly distributed and concentrated in the breast milk which contains approximately three times the concentration of nicotine compared with the maternal serum [12]. The half-life of nicotine in breast milk is 97 ± 20 min [13] indicating that recency of maternal exposure to nicotine impacts the continued delivery of nicotine to the neonate via this route.

#### 1.1.2. Nicotine General Actions

The effects of nicotine exposure are biphasic, individually variable, and influenced by age, gender, pregnancy, and genetics [4]. Additional factors include: The time of day, dose, formulation, administration method, time since last exposure, cumulative nicotine level, and the presence of medications and foods which can act as inhibitors and inducers [4]. At the macro level, nicotine is a vaso-constrictor with the consequence in pregnancy of decreased placental perfusion, resulting in decreased fetal oxygenation, leading to decreased birth weight, and increased relative risks of preterm birth, and fetal and neonatal morbidity and mortality [14,15].

At the micro level, nicotine mimics the action of acetylcholine and stimulates both the sympathetic and parasympathetic branches of the autonomic nervous system and exerts and acts as an agonist at nicotinic acetylcholine receptors (nAChRs) in the central and peripheral nervous systems, muscle and many other tissues [16]. In fetal development, nAChRs emerge early [17] and begin to form connections, with construction of axons and synaptic connections continuing after birth and into childhood and adolescence [18]. Placental transfer of nicotine stimulates nAChRs in the developing foetus [19], albeit in immature physiology, and results in neuroadaptation and nicotine tolerance which produces changes in both nicotinic receptors and neural plasticity [18]. This results in accelerated cell development relative to tissue and organ age, that is, there are fewer cells correctly developed for their stage and age [19] and a deficit in the number of neurons in the fetal/neonatal brain [20]. In addition, there is synaptic level damage to the respiratory center and the micro level neuro-teratogenic nAChR effects from nicotine occur at an exposure/dose threshold level below that which causes intrauterine growth retardation [21,22]. More recent research points to fetal nicotine exposure resulting in DNA methylation leading to genome and epigenetic changes and long-term sequelae in the children of smokers [23,24,25,26].

#### 1.1.3. Nicotine Metabolism and Excretion

The liver is the prime site for nicotine metabolism (via the CYP2A6 pathway) with the brain, kidneys, and lungs providing minor sites [27]. Cotinine is the main conversion output (70–80%) [28] with additional metabolites including nicotine-1′-N-oxide, nicotine glucuronide, and nornicotine [29]. Nicotine and its metabolites are rapidly excreted by the kidneys with the rate dependent upon the urinary pH, increased urine alkalinity decreases excretion.

As with nicotine actions, nicotine metabolism and excretion in both adults and neonates is impacted by individual variability including: Age, diet, hormonal fluctuations, gender, genetics, liver and renal function and medications [4]. In pregnancy, substantial induction of CYP2A6 activity occurs resulting in a 60% increase in plasma nicotine clearance, 140% increase in cotinine clearance, and a reduction in the half-life of cotinine of nearly 50% (down to nine hours) [30]. Consequently, findings of lower cotinine concentrations in pregnancy compared to pre-pregnancy or post-pregnancy do not necessarily indicate lesser exposure to nicotine but instead may indicate more rapid metabolism of nicotine. In neonates, differences in nicotine distribution and CYP2A6 activity result in a decreased ability to metabolize nicotine compared to adults, subsequently, the plasma nicotine half-life is three to four times longer (11.2 h) while the cotinine elimination half-life is similar to that of adults at 16.3 h [31].

#### 1.1.4. Tobacco and Nicotine Biochemical Analysis

The biological transit of tobacco biochemicals from maternal cigarette and ST exposure through to fetal exposure, and then to maternal and neonatal tobacco and nicotine excretion can be measured in maternal, placental and neonatal biological samples [4,12,29,31,32,33,34,35]. Physiologically, the dynamic nature of nicotine metabolism creates large plasma nicotine fluctuations, which pose a challenge for intra- and inter-personal comparisons of exposure, consumption, and effect. However, cotinine is relatively pharmacologically inactive and has a half-life of 17 h and serum concentrations 10-fold higher than nicotine, therefore, it is a useful biological indicator of nicotine use [36,37]. At approximately 16 h, cotinine begins to be metabolized to trans-3′-hydroxycotinine, nornicotine, nicotine glucuronide, and nicotine-N-oxide [29]. Given the dynamic nature of tobacco and nicotine metabolism and the fluctuations in metabolite concentrations, assessment of an individual’s tobacco and nicotine exposure can be achieved by summing the concentrations of nicotine and its metabolites in each sample [38]. In addition, tobacco itself produces other related alkaloids including anabasine and anatabine, the presence of which can be used to infer tobacco use [39].

### 1.2. Tobacco Exposure and Adverse Pregnancy Outcomes

#### 1.2.1. Maternal Smoked Tobacco Exposure and Adverse Pregnancy Outcomes

The initial scientific report that first associated maternal tobacco smoking with unfavorable birth outcomes was published in 1957 [40]. Extensive and intensive research over the ensuing 60 years has established a range of predictable adverse pregnancy outcomes in the presence of maternal smoked tobacco exposure [41,42,43] with newer research demonstrating increased pro-inflammatory responses to fetal nicotine exposure that persist through early development and adulthood [44,45]. Table 1 lists the more significant of the evidenced maternal, fetal, and childhood outcomes.

#### 1.2.2. Maternal Smokeless Tobacco Exposure and Adverse Pregnancy Outcomes

The 2015 Global Adult Tobacco Survey [51] represents three billion of the five billion worldwide adult population. The survey, while not measuring ST use in all countries, reported an estimated 248 million people used ST. In some populations, female ST use is high, for example, in Bangladesh 28% of women (13 million), and in India 18% of women (71 million) use ST compared with smoking in those same countries (1% = one million women, and 2% = seven million women, respectively). However, in that survey, the number of women who used tobacco in pregnancy was not reported. In the Demographic and Health Surveys data from 54 low-income countries, a wide variation in ST use in pregnancy was reported, with higher rates occurring in the South Asian countries (India 7.2%) and Africa (Madagascar 11.8%) [52].

An emerging focus on tobacco and nicotine delivered by non-smoking methods is seen in more recent literature. In 2010, England et al. [53] provided a descriptive summary of the adverse pregnancy outcomes associated with nicotine administered by an array of non-smoking methods. The summary included a synopsis of the literature around maternal oro-nasal ST. Four years later, a detailed integrative review of the literature [54] evaluating the effects of maternal oro-nasal ST use in pregnancy was conducted. That report indicated that maternal ST use was not safe for mother or foetus. That report was followed up by England et al. [18] with a synthesis of the theoretical and epidemiological literature around emerging tobacco and nicotine products in relation to pregnancy, childhood, and adolescent exposure.

#### 1.2.3. Maternal Smoking and Australian Birth Outcomes

In Australia, cigarette use during pregnancy is recognized as a specific adverse risk factor. As such, the Australian Perinatal National Minimum Data set [55] directs the capture and reporting of maternal smoking information on expectant mothers’ Perinatal Data Collection report. Other methods of tobacco or nicotine maternal use or exposure are not captured. The following specific yes/no questions are completed by the midwife: “*During the first 20 weeks…did the mother smoke at all?”*, and “*During the second 20 weeks…did the mother smoke at all?”* If the mother answers “*yes*”, the midwife is directed to quantify the number of cigarettes smoked per day, and to indicate if smoking cessation advice has been provided.

In the Australian Northern Territory (NT) Mothers and Babies 2015 report [56], the self-reported smoking rate for Indigenous mothers in the first 20 weeks of pregnancy was 48%. Conversely, a lower rate of 26% was recorded for Indigenous expectant mothers in the “Alice Springs Rural district” of the NT. “This pattern of lower smoking rates in Central Australia is consistent with data from previous years” and is surmised by the NT reporters to be related to “the local practice of chewing tobacco (pituri) in that region” [56].

### 1.3. Chewing Tobacco Use in Central Australia

In central Australia, *Nicotiana* spp. is commonly used as chewing tobacco by Aboriginal women [57]. The *Nicotiana* spp. are collectively known by a variety of names, one of which is pituri. The pituri is mixed in the hand with wood ash prior to brief mastication. The subsequent “quid” is held for extended periods of time in the lower and inner side of the lip, or between the teeth and the cheek. When not positioned in the oral region, the moist quid is usually stored on the skin behind the ear—possibly providing a transdermal nicotine administration route. Botanical analysis of Australian *Nicotiana* spp. demonstrates nicotine concentrations as high as 8 mg/g [3], and analysis of commonly used wood ash demonstrates high alkalinity (pH’s > 12.0 [57]) indicative of their ability to assist extraction and absorption of nicotine and related compounds during chewing [58].

The practice of chewing pituri continues throughout pregnancy, childbirth, and lactation [57]. In 1986, the only large tobacco-use survey of this area reported “most [Aboriginal] women chew in central Australia” [59]. Contemporaneous observation of Aboriginal women birthing at the Alice Springs Hospital (ASH) indicates that 33% have visible oral quids. Given that the pituri may be positioned in and on various body sites, observation of visible oral quids likely results in an underestimation.

### 1.4. Study Aim and Research Questions

The complex physiological response to nicotine exposure including inflammation, oxidative stress, and the role of the endoplasmic reticulum and protein synthesis are areas of current research [15,60,61,62]. These intricate scientific questions and research are of fundamental importance in understanding the role of nicotine exposure by means other than combusted tobacco inhalation in adverse pregnancy outcomes [63,64,65,66,67]. Nevertheless, in the context of smokeless tobacco use in central Australia, there has been no research examining the effects of pituri on general health outcomes, nor on pregnancy and birth outcomes. The aim of this research is to measure and compare maternal and perinatal outcomes between central Australian Aboriginal women with different levels of self-reported tobacco exposure. The research questions are:What are the biochemical concentrations of tobacco, nicotine, and their metabolites in a range of maternal and neonatal biological samples from mothers with differing levels of self-reported tobacco use?Does maternal pituri chewing impact maternal and perinatal outcomes, and if so, what is the significance of that impact in comparison to smoking or non-tobacco use in pregnancy?

### 1.5. Research Setting and Considerations

In this research, “central Australia” is the geographical area serviced by ASH [68] which includes the area covered by the Ngaanyatjarra Pitjantjatjara Yankunytjatjara Women’s Council [69], Tennant Creek, the Barkly Tablelands, the Central Desert Shire, and the MacDonnell Shire of the Northern Territory (Figure 2 [70]). As ASH is the primary location where central Australian Aboriginal women birth, it was chosen as the research site.

This research centers upon the confronting issue of tobacco use in pregnancy and exists alongside the ethical complexities of the population’s cultural, linguistic, and geographical diversity [71,72]. The development of relationships conducive to this research occurred over a period of years, and included working closely with several Aboriginal organizations, and in particular the Ngaanyatjarra Pitjantjatjara Yankunytjatjara Women’s Council (NPYWC). From the outset, the NPYWC Directors were adamant that pituri was not a tobacco plant. The research potentially questioned the knowledge foundations of Aboriginal women, creating a dilemma for the Directors around the possible use of tobacco in pregnancy [57].

Several years of engagement with the NPYWC enabled an intricate, negotiated research space to evolve where conversations around pituri use commenced, and substantial consultation was undertaken related to the research, its ethical issues, its processes, and logistics. As such, the research processes were structured around a co-operative investigation that connected the skills and knowledge of central Australian Aboriginal women with those of academics and clinicians. Intentionally in terms of ethical considerations, the recruitment processes and the conduct of the research were informed and guided by senior central Australian Aboriginal women to ensure the research procedures supported participant privacy and informed consent, and importantly, did not impinge on cultural congruence or safety matters.

The research was approved by The University of Queensland Human Research Ethics Committee (HREC #2010000548) and the Central Australian HREC (#2010.06.04). Following ethics approval, another 12-months was invested in developing participant recruitment materials prior to final stakeholder endorsement of the project.

## 2. Methods and Design

### 2.1. Study Design

The study utilized an observational design. Potential participants were invited to participate if they self-identified as Aboriginal, were at least 18 years old, were greater than 28 weeks gestation with a singleton pregnancy, and were planning to birth at ASH.

#### Sampling, Recruitment, and Consent

A pragmatic sampling approach was used to sample the participants. Hardon et al. [73] define pragmatic sampling as a non-probability approach where the constraints of the environment and resources are accounted for in the access to, and availability of, participants. Potential participants were identified when they attended an antenatal appointment or care at ASH. Potential participants were identified by researchers who were specifically trained in the use of the research tools and skilled in communicating sensitively with central Australian Aboriginal women. Participation was not sought based on known tobacco use status.

Recruitment videos were constructed by senior Aboriginal women from central Australia in the Aboriginal languages of Arrernte, Pitjantjatjara and Warlpiri, and in English, and were shown when potential participants attended at ASH for antenatal appointments. The videos detailed the research requirements, including the collection of biological samples from both the mother and neonate. Potential participants were provided with the opportunity to ask questions, and an interpreter was available when required. Only after the appropriate language video had been viewed, and the potential participant indicated their desire to participate was written consent obtained from participants for both their own and their neonates’ biological sample collection. Participants were informed that participation was voluntary and that they could refuse to participate or opt out of any specific component of the research without withdrawing completely from the study. Participants were specifically informed that any decision they made would have no effect on their pregnancy care or the care of their neonate.

The trained researchers were mindful of the non-verbal cues given by central Australian Aboriginal women when they did not wish to engage in non-clinically related dialogue, and these expectant mothers were not approached. In addition, potential participants were not approached if supporting community members were in attendance in order to avoid “shaming” the potential participant or community members and to mitigate breaching the potential participant’s privacy and confidentiality. For Aboriginal people, Blagg [74] explains that “shame” is a multifaceted, cultural paradigm which is “bound up with anxieties about a loss of social status, a feared mortification of a public self”. Being favorably or unfavorably singled out creates anxiety and uncertainty and Aboriginal people will strive to avoid being “shamed” [74].

The approaches were only made when there was time available to comprehensively explain the research questions, the research aims, participant requirements, undertake the recruitment procedures and obtain consent, and complete the maternal interview without haste. When a female central Australian Aboriginal language interpreter was required and not available, the potential participant was not approached. Additionally, potential participants were not approached if they were in established labor. While this recruitment strategy severely limited the number of potential approaches to approximately two-three per month, the approach did not favor or bias any tobacco-use group.

The sampling process included participants with known social and physiological predictors for adverse maternal and perinatal outcomes. These non-exclusion criteria were informed by the endemic rate of adverse maternal and perinatal outcomes predictors in the study population—exclusion based on endemic risk factors would have precluded many/most of the study population from the research. Social predictors included: Being Indigenous, geographic isolation, limited access to health care, lower levels of education, poor and overcrowded housing, high levels of domestic violence, limited access to antenatal education, and poverty [75,76]. Physiological predictors included: Poor nutrition, anemia, hypertension, diabetes, tobacco use, and alcohol use [68]. The significance of the different predictors was considered in the analysis.

There was one exclusion criterion—self-reported dual pituri and cigarette use. This exclusion was based on the inability of the maternal and neonatal outcome variables and biochemical measures to categorically discriminate between the maternal and neonatal effects and biochemical concentrations of tobacco and nicotine absorbed through the respiratory tract, and the maternal and neonatal effects and biochemical concentrations of tobacco and nicotine absorbed through the oral and transdermal routes [77].

*Sample size*. Based on the homogeneity of this maternal birthing population and the rate of observed oral pituri quids, at 80% power, with a 0.2 effect size and an alpha level of 0.05, at least 60 participants were required to detect significant differences between the group’s tobacco, nicotine and metabolite concentrations [78].

## 3. Procedure

### 3.1. Data Collection

In this research, the analytical inquiry required the collection and analysis of the maternal and perinatal data and biological samples. The aim was to consider the use of maternal smokeless tobacco as a possible causal factor in adverse maternal and perinatal outcomes. As such, epidemiological research identifies a range of variables—in addition to tobacco exposure—that influence maternal and perinatal outcomes [79,80,81,82,83,84,85]. These indicative and influencing variables are reflected and measured in the capture of both Australian maternal and perinatal data [86,87,88] and global maternal and perinatal data [89]. However, within the study population, there are additional extraneous and confounding variables known to impact pregnancy outcomes. As there had been no previous research examining the use of Australian *Nicotiana* spp. by Australian Indigenous people, these variables, as well as those in the standard measure of maternal and perinatal outcomes, and the biological samples were collected in the following manner:(1)A maternal face-to-face interview was conducted at the time of enrolment and consisted of 23 semi-structured questions recording information not collected elsewhere, for example: Language group, skin name, employment and income, housing situation, community of origin and school leaving level, in addition to lifestyle factors including alcohol and tobacco use. The participants were clearly informed that the research was exploring the effect of tobacco and pituri use in pregnancy. The interviewer read each question to the maternal participant, and their responses were documented verbatim on the interview tool. “Yarning” [90,91] was used to navigate through the data elements of the maternal interview. Yarning was conducted in a conversational style which enabled the participant to elaborate on responses as they chose. The tool was designed to enable the participant to share their ethnobotanical knowledge of pituri preparation and use as chewing tobacco. This aspect of the data collection informed the development of a derivative project which examined Aboriginal pharmacological knowledge of pituri and its use [57].(2)The second data collection tool obtained maternal and perinatal data from each participant’s electronic medical record. In Australia, the capture of mandatory information pertaining to each Australian pregnancy and birth is directed by the Australian National Minimum Perinatal Data Set [92]. In the NT, this information is collected and electronically stored as part of the NT Perinatal Data report within CARESYS^®^, the electronic medical record [93], hereafter referred to as the CARESYS^®^ data. The CARESYS^®^ data records 80 primary and 40 secondary maternal and perinatal variables. The CARESYS^®^ information is progressively entered by attending health care professionals throughout the pregnancy. For this research, a complete Perinatal Data record was printed from CARESYS^®^ following the participant’s hospital discharge after birthing. Table 2 lists the maternal interview and the CARESYS^®^ data variables and their operational definitions collected and used in this research.(3)The third data collection involved the collection of biological samples from the participant and their neonate. The objective of this was to quantify the level of tobacco and nicotine and metabolite concentrations in maternal and perinatal samples from different self-reported tobacco exposures as a question of possible causality in adverse maternal and perinatal outcomes. As such, epidemiological research [30,94,95] demonstrates that the transit of biochemical tobacco and nicotine from maternal exposure through to fetal exposure, and then to maternal and neonatal excretion [4,12,29,31,32,33,34,35] can be measured in maternal, placental and neonatal biological samples. Informed by the literature, Table 3 details the biological samples and their rationale for collection in response to this question. The biological samples and their collection procedures are standard clinical procedures and required no additional staff training and minimal clinical resources. The samples were collected at the time of enrolment (maternal venous blood, urine, and hair) and as they became available during the study and were identified with a unique participant identifier before being transferred to the hospital pathology for storage in a −80 °C freezer.

### 3.2. Data Processing

The variables listed in Table 2 were extracted from the participants’ maternal interview data and Excel^®^ reports and transcribed into an Excel^®^ database designed for this research. The participants self-reported tobacco use from the maternal interview stratified the participant as one of the following: (1) Pituri user, (2) smoker, or (3) non-tobacco user. Where two sources of the same information were available for cigarette and alcohol use, for example, the medical record and the maternal interview, the maternal interview (i.e., what the mother self-reported at the time of enrolment), was used. The data were cleaned and verified by rechecking each entry on three occasions. Data were then imported into SPSS^®^, and validation of the data accuracy following transfer was conducted by matching each entry to the Excel^®^ record.

### 3.3. Biological Sample Processing

Biological samples were periodically transferred to The University of Queensland’s Pharmacy Australia Centre of Excellence (PACE) for analysis. Biological samples were stored at PACE in a dedicated freezer at −80 °C until analysis. The scientists were blinded to the self-reported tobacco-use status of the participant and analyzed the biological samples according to the following procedures:

#### 3.3.1. Reagent Setup

Specific alkaloids and deuterated internal standards were obtained from Toronto Research Chemicals (Toronto, ON, Canada): (R,S)-anabasine and (R,S)-anabasine-2,4,5,6-d4, (R,S)-anatabine and (R,S)-anatabine-2,4,5,6-d4, (−)-cotinine, (+/−)-nornicotine, nicotine-N-β-glucuronide, trans-3′-hydroxycotinine. (−)-nicotine, (+/−)-nicotine-d4, (+/−)-cotinine-d3 and (+/−)-nornicotine-d4 were purchased from Sigma (St. Louis, MO, USA). Nicotine-N-β-glucuronide-methyl-d3, trans-3′-hydroxycotinine-d3, (1′S, 2′S)-nicotine-1′-oxide and (+/−)-trans-nicotine-1′-oxide-methyl-d3 were obtained from Santa Cruz Biotechnology (Santa Cruz, CA, USA). The purity of all the commercially available reference compounds was assessed. Methanol and acetonitrile were purchased from Merck (Darmstadt, Germany) and ammonium hydroxide from Sigma. Ultra-pure water was prepared by a Millipore Milli-Q system (Milford, MA, USA). Stock solution for the alkaloid standards was prepared in methanol at a concentration of 1 mg/mL and was stored at −80 °C. Working solutions were prepared by serial dilution from the stock solutions and stored at −20 °C until required for an analytical run. Internal standard mixture was prepared by addition of the eight-deuterated internal standards at 1 µg/mL concentration in methanol and stored at −20 °C.

#### 3.3.2. Sample Preparation

The native fluids (urine, serum and colostrum and/or breast milk) were thawed and centrifuged at 12,000× *g* for 5 min to remove solid impurities.

Urine. An aliquot of urine (20 µL) was spiked with 5 µL of 1 µg/mL internal standard mixture and basified with 1 mL of 0.15 M aqueous-ammonium hydroxide solution. Urine samples were prepared and extracted using a modification of a published method [111]. Following brief vortex, the solution was loaded onto the conditioned (1 mL methanol) and equilibrated (1 mL 0.15 M ammonium hydroxide) SampliQ Polymer SCX (30 mg, 1 mL, Agilent Technologies). The loaded cartridge was washed with 1 mL of 2% formic acid and eluted with 2 mL of methanol followed by 2 mL of 3% ammonium hydroxide in methanol. The eluent collected in a 16 × 100 mm glass tube was evaporated in a Pierce Reacti-Therm III heating module (Rockford, IL, USA) to approximately 5–10 µL residual volume to avoid loss of volatile compounds such as cotinine and 3′-hydroxycotinine. Subsequently, the residue was reconstituted in 100 µL of methanol and re-centrifuged in 1.5 mL of tube for 5 min at 12,000 rpm and 1 µL of the solution and injected into LC-MS/MS system.

Plasma and colostrum and/or breast milk. An aliquot of plasma (50 µL) or colostrum and/or breast milk (50 µL) was spiked with 5 µL of 1 µg/mL internal standard mixture, and then 100 µL of acetonitrile was added to the solution to aid in matrix clean-up. Following vortex-mixing and 10 min of 12,000 rpm centrifugation, the supernatant was transferred to a 16 × 100 mm glass tube and evaporated by nitrogen gas flow at 40 °C for 10 min in the Pierce Reacti Therm III heating module. The residue was reconstituted in 100 µL of methanol, re-centrifuged in 1.5 mL tube for 5 min at 12,000 rpm and 1 µL of the solution was injected into LC-MS/MS system.

#### 3.3.3. LC-MS/MS Analysis

Liquid chromatographic separation was achieved on a Poroshell 120 HILIC column (2.1 × 150 mm, 2.7 µm) with an 1290 Infinity inline filter (0.3 µm, Agilent, Santa Clara, CA, USA) with a gradient system consisting of 10 mM ammonium formate with 0.1% formic acid (pH 3.2, A) and methanol with 0.1% formic acid (pH 3.2, B) at a flow rate of 200 µL/min. The initial mobile phase condition was 5% A which increased linearly to 30% for 3 min followed by an increase to 40% for 5 min, then decreased back to the initial mobile phase condition of 5% A within 1 min and re-equilibrated for 30 min.

Chromatographic analysis for urine and plasma was carried out using a hybrid triple quadruple mass spectrometer API 3000 equipped with a turbo ion spray source, operating in positive ESI mode supported by Analyst software (AB Sciex, Darmstadt, Germany). For the MS/MS scan, the precursor ions were selected and the product ions generated using different collision energy, the most intense product ion was selected for the multiple reaction monitoring mode (MRM) scan. All MRM transitions were monitored with a unit mass resolution for both precursor and product ions. The turbo ion spray source was maintained using the flowing setting: Ion spray voltage = 5000 V, turboprobe temperature = 300 °C, heater gas flow 7 L/min, nebulizer setting (GAS1) = 15.

Chromatographic analysis for colostrum and/or breast milk was conducted using an Agilent 6460 Triple Quadrupole tandem mass spectrometer equipped with a Jet Stream source as the detector with Mass Hunter Workstation software (Agilent Technologies, Santa Clara, CA, USA). The mass spectrometer was operated in electrospray positive mode using MRM data acquisition that was optimized to the highest possible sensitivity for each alkaloid. The following ESI conditions were applied: gas flow = 5 L/min, sheath gas flow = 12 L/min, gas temperature = 300 °C and sheath gas temperature = 250 °C.

#### 3.3.4. Method Validation

The linearity of the method was assessed using six serial dilutions from the alkaloid stock solution and internal standard by linear regression analysis. Each dilution was used to construct the calibration plots by plotting the peak area ratio of each alkaloid and its deuterated internal standard versus the concentration of alkaloid. The limits of detection (LOD) and limits of quantitation (LOQ) for each alkaloid were determined based on the concentrations (as peak heights) corresponding to 3× noise and 10× noise respectively. To assess the precision of the method, 10 individual aliquots of the fluids samples were evaluated followed by the calculation of relative standard deviation (RSD) for the peak area ratio of each alkaloid and its deuterated internal standard. Recovery was calculated as the measured concentrations divided by the expected concentrations by using six aliquots each of spiked and non-spiked fluid samples. Each aliquot of milk and serum was spiked to give an added concentration similar to the mid-point of the calibration range.

The findings were reported by the unique participant identifier on an Excel^®^ spreadsheet. Following data validation, the unique identifier was matched to the participants’ maternal results and the results uploaded into SPSS^®^ version 20.0 (IBM^®^ SPSS^®^ Statistics 20) for data analysis. 

### 3.4. Data Analysis

For each biological sample, the nicotine and its metabolites cotinine, nicotine glucuronide, nicotine-N-oxide, nornicotine and trans-3′-hydroxycotinine were combined to provide a measure of nicotine exposure [38]. Concentrations of anabasine and anatabine were used for comparison with self-reported tobacco use to indicate participant exposure (smoked or chewed) to tobacco plant material [39].

The association between tobacco use and maternal and neonatal outcomes was investigated and reported using the STROBE guidelines [112]. Missing data and outliers are reported but are excluded from analysis. Outliers are identified by the use of boxplots and are excluded from data analysis based on established procedures [113]. Results are reported with the level of significance at alpha 0.05 and accompanied by 95% confidence intervals (95% CI).

Descriptive and comparative analyses were undertaken using self-reported tobacco use as the independent variable. Means ± standard deviations or proportions are reported for in-group dispersion and central tendency where applicable. Continuous variables are tested for statistically significant differences between exposure groups using t-test and ANOVA, followed by appropriate post-hoc tests when required. Categorical variables are analyzed by chi-square with appropriate post-hoc tests when required.

Multivariate logistic and linear regression, with adjustment based on previous literature examining tobacco use and pregnancy outcomes [9,19,43,96,114,115,116,117,118,119], was conducted. Variables were measured as continuous or categorical (Table 2) and fitted as covariates in that manner in the regression models. Variables with a *p* < 2 in the regression model were accepted for covariate interaction inclusion. The variables broadly categorized as follows were used to investigate predictor and/or covariate and/or outcome variables:Demographic variables (age, residential address, education level)Lifestyle factors (alcohol and tobacco use)Past and current medical history (cardiac, hypertension, diabetes and renal disease)Pregnancy-related factors (parity, gravida, elevated glucose, hypertension, STI, UTI, anemia, number of antenatal visits, placental weight and size)Labor and birthing factors (LUSCS, meconium staining, post-partum hemorrhage)Birth outcome variables (gestational length, birth weight, gender, admission to SCN and APGAR score)

## 4. Discussion

This research design is based on the hypothesis that maternal exposure to tobacco and nicotine through non-combusted means, in this instance, the use of pituri as chewing tobacco, will result in adverse maternal and neonatal outcomes. This hypothesis is founded upon the intensive research conducted over the past 60 years examining the botanical-pharmacological-biochemical pharmacokinetic and pharmacodynamic principles around tobacco and nicotine exposure which have established a causal link between both the maternal use of burnt tobacco and exposure to the products of its combustion (second-hand smoke) and adverse maternal and perinatal outcomes.

The study consists of various strengths and limitations. The major strength is the homogenous population cohort, the inclusion of three tobacco exposure groups, with the non-tobacco users acting as controls, and the comprehensive sample collection planned to allow investigation of nicotine exposure of both mother and neonate. The inclusion of participants with known social and physiological predictors for adverse maternal and perinatal outcomes enables the findings to be considered within the context of the study population and assists with the generalisability of the findings to wider worldwide populations where smokeless or chewed tobacco is used in pregnancy. The limitations include the observational nature of the study with the enrolment of participants using a pragmatic approach and only after 28 weeks gestation, thus the once-only collection of biological samples after this date impacts the question of maternal and fetal nicotine exposure prior to enrolment and in addition, first and second-trimester adverse pregnancy outcomes.

## Figures and Tables

**Figure 1 mps-02-00047-f001:**
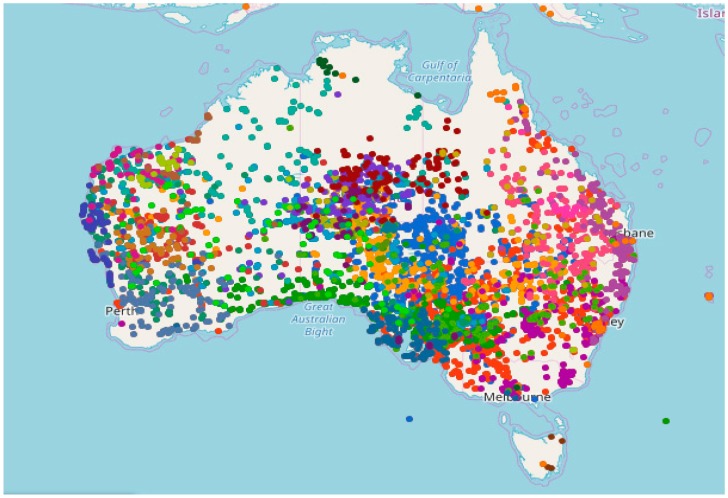
General dispersal of Australian *Nicotiana* spp. [2].

**Figure 2 mps-02-00047-f002:**
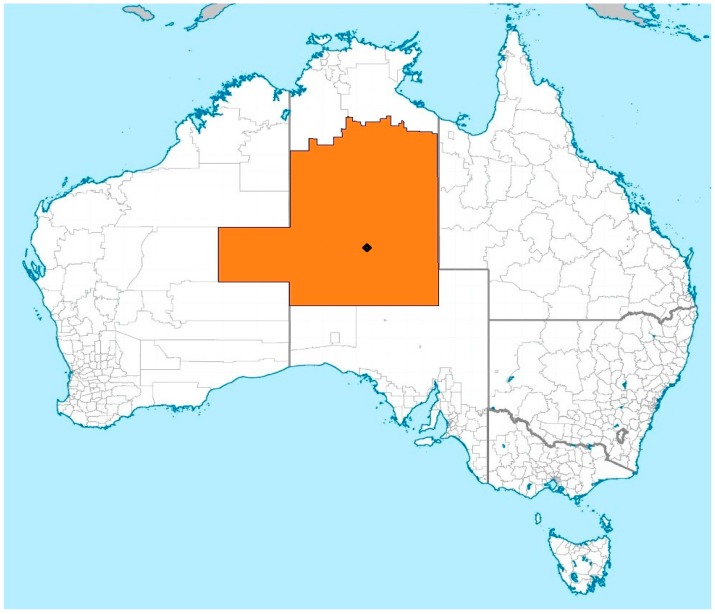
In this research, the ochre coloured area represents the area serviced by the Alice Springs Hospital (marked as ♦). Map based on the Local Government Areas [70].

**Table 1 mps-02-00047-t001:** Increased relative risk outcomes from maternal exposure to combusted tobacco [41,46,47,48,49,50].

Maternal Outcomes	Fetal/Neonatal Outcomes	Childhood/Adolescent Outcomes
Antepartum hemorrhage Placental previa Placental abruption	Low birthweight Small for Gestational Age (SGA) Intrauterine Growth Retardation (IUGR)	Obesity Metabolic disorders
Miscarriage Ectopic pregnancy	Preterm labor and delivery	Type 2 diabetes
Placental changes, hypertrophy, calcification	Stillbirth, neonatal death, SIDS	Hypertension
Venous thrombosis, Pulmonary embolism	Microcephalus Cleft defects Clubfoot deformities	Neurobehavioural changes

**Table 2 mps-02-00047-t002:** Maternal Interview and Exposure Variables and Maternal and Perinatal Outcome Variables.

Variable	Operational Definition of Variable	Measurement Scale
Aboriginal language group	Central Australian Aboriginal language group	Nominal
Access to secondary health care	Maternal residential distance from Alice Springs	Continuous: km
Admission to Special Care Nursery (SCN)	Any admission of the neonate to any special care nursery following birth	Dichotomous: yes/no
Age	Maternal age at last birthday	Continuous: years
Alcohol use	Maternal interview, self-report of any alcohol use in this pregnancy	Dichotomous: yes/no
Anemia	<110 g/L hemoglobin in venous blood	Dichotomous: yes/no
Antepartum hemorrhage (APH)	Any antepartum haemorrhage	Dichotomous: yes/no
Apgar 1, 5, 10 min	Neonatal score of 0, 1 or 2 for: Heart rate, breathing, color, muscle tone and reflex irritability. Total 0–10.	Ordinal
Augmentation	The stimulation of ineffective uterine contractions after the onset of labor, to manage labor dystocia.	Dichotomous: yes/no
Birthing method	Method of birth	Nominal: SVB, LUSCS, forceps, ventouse
Body mass index (BMI)	Ratio between weight (kg) and height (cm) as measured by weight divided by height squared = kg/m^2^	Ordinal: underweight, obese normal, overweight
Born before arrival (BBA)	Birth that occurs before arrival at the Alice Springs Hospital	Dichotomous: yes/no
Cardiac disease	Health practitioner diagnosis of any maternal cardiac disease	Dichotomous: yes/no
Cigarette use	Maternal interview, self-reported use of cigarettes in this pregnancy	Dichotomous: yes/no
Diabetes Mellitus	Health practitioner diagnosis of pre-gestational diabetes	Dichotomous: yes/no
Duration of labor	Duration from onset of established labor to complete birth of neonate	Continuous: hours
Education level	Self-reported school leaving grade	Continuous: grade
Variable	Operational definition of variable	Measurement scale
Elevated glucose	Maternal elevated glucose where inadequate identification of pre-gestational diabetes or gestational diabetes status exists	Dichotomous: yes/no
Episiotomy	Perineal incision to facilitate birth of neonate	Dichotomous: yes/no
Forceps	Instrumental delivery of the neonate via the vagina	Dichotomous: yes/no
Gender	Gender of the neonate	Dichotomous: male/female
Gestation at 1st antenatal visit	Time since last menstrual period and attendance at 1st antenatal visit	Continuous: completed weeks
Gestation at 1st ultrasound	Time since last menstrual period and attendance at 1st ultrasound	Continuous: completed weeks
Gestational diabetes mellitus (GDM)	Health practitioner diagnosis of diabetes that develops during pregnancy	Dichotomous: yes/no
Gestational length	Time since last menstrual period and birth of neonate	Continuous: completed weeks
Gravida	Number of times a woman has been pregnant regardless of whether the pregnancies result in a live birth	Discrete: number
Head circumference	Neonatal head circumference at birth	Continuous: cm
Housing situation	Self-report of number of residents that live with participant	Discrete: number
Hypertension (pre-gestational)	Health practitioner diagnosis of pre-gestational hypertension	Dichotomous: yes/no
Hypertension	Maternal elevated blood pressure where inadequate identification of pre-gestational hypertension or pregnancy-induced hypertension status exists	Dichotomous: yes/no
Income	Self-report of weekly income	Continuous: Australian dollars
Induction and indicator	The purposeful stimulation of uterine contractions for the purpose of accomplishing delivery, prior to the natural onset of labor	Dichotomous: yes/no
Labor complications	Health practitioner diagnosis of labor complications	Dichotomous: yes/no
Livebirth	Neonatal outcome following the complete expulsion or extraction from its mother which after separation, shows signs of life	Dichotomous: yes/no
Lower Uterine Segment Caesarean Section (LUSCS)	Operative delivery of the neonate from the uterus via the abdomen	Dichotomous: yes/no
Meconium stained liquor	Presence of meconium in liquor	Dichotomous: yes/no
Membranes complete	Presence of complete membranes	Dichotomous: yes/no
Neonatal abnormalities	Presence of any neonatal abnormalities	Dichotomous: yes/no
Neonatal body length	Neonatal body length at birth	Continuous: cm
Number of antenatal visits	Number of antenatal visits recorded in the perinatal record following birth	Discrete: number
Number of cord vessels	Visual inspection of cord after separation from neonate	Discrete: number
Parity	Number of previous pregnancies resulting in live births or stillbirths, excluding the current pregnancy	Discrete: number
Pituri use	Maternal interview, self-reported use of pituri in this pregnancy	Dichotomous: yes/no
Placenta complete	Presence of complete placenta	Dichotomous: yes/no
Placental abruption	Placental separation prior to birth of the neonate	Dichotomous: yes/no
Placental lie	Relationship of the maternal axis to the fetal axis	Nominal: longitudinal, transverse, oblique
Placental previa	Placental lie across the cervical os	Dichotomous: yes/no
Placental size	Diameter of placenta at the two widest points in cm. Result multiplied together to find area (cm^2^)	Continuous: cm^2^
Placental weight	Weight of the placenta following drainage of blood	Continuous: grams
Post-partum hemorrhage	>500 mL blood loss in first 24 h post birth	Dichotomous: yes/no
Pre-eclampsia – eclampsia	Hypertension, oedema and proteinuria during pregnancy	Dichotomous: yes/no
Pregnancy complications	Health practitioner diagnosis of pregnancy complications	Dichotomous: yes/no
Pregnancy-induced hypertension	Health practitioner diagnosis of hypertension that develops during pregnancy	Dichotomous: yes/no
Premature rupture of membrane	Rupture of membranes <37 weeks gestation	Dichotomous: yes/no
Presentation	Part of the neonate presenting at the superior aperture of the maternal pelvis	Nominal: cephalic, breech, shoulder
Previous adverse obstetric history	Health practitioner diagnosis of adverse obstetric history	Dichotomous: yes/no
Race	Self-report of race	Nominal
Rubella immune status	Rubella IgG antibody level >10 IU/mL	Dichotomous: yes/no
Sexually transmitted infection	Health practitioner diagnosis of sexually transmitted infection	Dichotomous: yes/no
Significant adverse medical history	Health practitioner diagnosis of any significant adverse medical history	Dichotomous: yes/no
Spontaneous vaginal birth (SVB)	Unassisted vaginal birth	Dichotomous: yes/no
Stillbirth	Neonate with no signs of life following the complete expulsion or extraction from its mother	Dichotomous: yes/no
Third stage method (active)	Method of delivery of placenta and membranes	Dichotomous: yes/no
Urinary tract infection	Health practitioner diagnosis of any urinary tract infection	Dichotomous: yes/no
Ventouse	Assisted birth using a suction cap applied to the neonate’s head.	Dichotomous: yes/no

**Table 3 mps-02-00047-t003:** Biological sample, rationale for collection and collection processes.

Biological Sample	Rationale for Collection	Collection Process	Collection Method
Maternal venous blood	Indicates recency of maternal nicotine exposure [94].	Maternal plasma collected concurrently with other plasma collections in order to minimize participant discomfort.	FBC for Hb. If recently collected do not repeat. Standard pink top tube. U&E for standard biochemical measures. If recently collected do not repeat. Standard green top tube. Tobacco, nicotine and its metabolites: 2 × 2 mL lithium heparin light green non-gel tubes.
Venous cord blood and Arterial cord blood	Indicates nicotine placental transfer and fetal exposure. Nicotine rapidly crosses the placental barrier with considerable amounts of nicotine occurring in the fetal blood of maternal smokers [96,97] and ST users [98]. By analyzing venous and arterial cord blood, a determination can be made of: (a) Fetal nicotine exposure (venous cord), and (b) fetal nicotine circulating levels (arterial cord).	Cord blood will be collected as per standard arterial and venous cord blood collection procedures following the complete expulsion of the placenta from the uterus and the complete separation of the placenta from the neonate.	Arterial: 2 × 2 mL lithium heparin light green non-gel tubes. Venous: 2 × 2 mL lithium heparin light green non-gel tubes.
Amniotic fluid	Amniotic fluid demonstrates fetal exposure to nicotine that penetrates through the amniotic membrane and fetal excretion (via the fetal kidneys and lungs) of nicotine and its metabolites into the amniotic fluid. Amniotic fluid concentrations are expected to be significantly higher than the umbilical arterial and venous concentrations as the foetus ingests, metabolizes and excretes and then re-ingests the amniotic fluid through the pregnancy [10,19,99,100].	A clean sample of amniotic fluid (visibly uncontaminated with maternal blood or meconium) will be obtained at lower uterine segment cesarean section (LUSCS).	10 mL, sterile yellow top collection jar.
Neonatal urine Day 1 Day 3	Fetal urinary concentrations of nicotine fluctuate dependent on recency of exposure, level of exposure, and metabolism rate and therefore are less indicative of long-term exposure than meconium. However, comparative analysis of Day 1 and Day 3 urine and to venous and arterial cord blood concentrations may demonstrate neonatal metabolism and excretion capacity following separation from the nicotine supply through the placenta [31]. If the neonate is breastfed and the mother is using tobacco in the early post-delivery period, colostrum may contain nicotine concentration at least equivalent to that of the breast milk therefore Day 3 urine testing will not provide an absolute indication of neonatal metabolism of nicotine following placental separation as they will have been exposed to nicotine through the colostrum and/or breast milk.	Neonatal urine collection bags will be placed on the neonate after birth and again on Day 3 and collected when available and uncontaminated with meconium, i.e., a clean sample.	Day 1: 2–3 mL, sterile yellow top collection jar Day 3: 2–3 mL, sterile yellow top collection jar
Meconium	Meconium is a fetal gut excretion product that begins to develop at about 12 weeks gestation and is generally not eliminated during the pregnancy. Meconium nicotine concentrations reflect fetal exposure throughout the second and third trimesters (i.e., longevity of exposure is demonstrated). Drug metabolite testing in meconium demonstrates high concentrations are detected in the meconium (100%) compared to urinary screens for the same drugs (37%) and that meconium testing has both high sensitivity and specificity [31,101,102].	Mothers will be encouraged to collect the neonatal meconium when it becomes available.	One scoop: Brown top sterile fecal collection jar, within first three days of birth
Colostrum and/or breast milk	Colostrum and breast milk is an excretion process and a possible route of post-birth nicotine exposure. The acidic milk compartments of the breast concentrate nicotine [12], with nicotine levels in breast milk reaching considerably higher levels than in the serum [103].	Maternal colostrum and/or breast milk will be concurrently collected with colostrum and/or breast milk expression in order to minimize participant discomfort.	2–3 mL, sterile yellow top collection jar
Maternal hair	Nicotine and its metabolites are deposited in hair from the time of exposure [104,105,106,107,108,109,110] indicating duration of exposure, changes in exposure during pregnancy are also indicated.	The maternal and neonatal hair samples will be collected from the nape of the neck, with mothers encouraged to obtain their neonates’ hair.	Several strands, sterile yellow top collection jar
Neonatal hair	Fetal hair begins to grow in the last three months of pregnancy, accumulating and concentrating cotinine and reflecting third-trimester exposure to nicotine [102,107]	As above	As above
Placenta	Placental size and weight indicative of neonatal perfusion.	________cm x ________cm __________grams	Measured across from the edge across the broadest sides and weight placenta in grams.

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
