# Peer review of "Tobacco and Pituri Use in Pregnancy: A Protocol for Measuring Maternal and Perinatal Exposure and Outcomes in Central Australian Aboriginal Women"

_mps, 2019, doi:10.3390/mps2020047_

Round 1
Reviewer 1 Report
The article is interesting and original.
The literature reported is comprehensive.
Here below some observations:
Line 64: I would add also novel tobacco products such as heat-not-burn tobacco products
Line 129: however, Line 131: however; there is too close repetition of the word however, I suggest to change the word.
I suggest to shift the considerations of lines 145 to 148, which are not related to the next sentence that is about maternal combusted tobacco exposure effects reported in table 1, and not to nicotine exposure by means other than combusted tobacco inhalation. This appears confusing.
So you could put the sentence 145-148 after the sentence "Table 1 lists the more 148 significant of the evidenced maternal, foetal and childhood outcomes.", or for example you can shift it later at the beginning of paragraph 1.4, to better introduce the nicotine exposure to non combusted products.
144: add period after [44,45]
Author Response
Reviewer One comments in black italics font.
The article is interesting and original.
Thankyou for this encouraging feedback.
The literature reported is comprehensive.
Thankyou for this encouraging feedback.
Here below some observations:
Line 64: I would add also novel tobacco products such as heat-not-burn tobacco products
Thankyou, tobacco product added to Line 64.
Line 129: however, Line 131: however; there is too close repetition of the word however, I suggest to change the word.
Thankyou, deleted second ‘however’.
I suggest to shift the considerations of lines 145 to 148, which are not related to the next sentence that is about maternal combusted tobacco exposure effects reported in table 1, and not to nicotine exposure by means other than combusted tobacco inhalation. This appears confusing.
So you could put the sentence 145-148 after the sentence "Table 1 lists the more 148 significant of the evidenced maternal, foetal and childhood outcomes.", or for example you can shift it later at the beginning of paragraph 1.4, to better introduce the nicotine exposure to non combusted products.
Thankyou, the paragraph has been moves to the beginning of 1.4.
144: add period after [44,45]
Thankyou, period added.

Reviewer 2 Report
The sample size calculations seem a bit vague. The investigators state that they wish to examine the associations between ST and adverse pregnancy outcomes; which outcomes and how will they be assessed? Do all births take place in hospital? Which outcomes has implications for the sample size calculations.
Further, do all women in this geographic area attend the antenatal clinics? If not, can a description be provided regarding the differences between those who do attend and those who do not? It is likely that more at risk women may not attend the clinics (in remote locations, etc).
Are there electronic records for births that do not take place in medical facilities?
What were the compliance rates for the biological samples, and was there any bias introduced by women who agreed vs those who did not?
When was breast milk obtained? Usually milk does not fully come in until approximately 5-7 days post birth, so collection would need to be done in the home.
It is likely that the control variables (i.e. confounders) will differ by outcome. Please specifiy which variables for which analysis using a DAG.
Author Response
Response to Review 2 comments (in black, italics font).
The sample size calculations seem a bit vague. The investigators state that they wish to examine the associations between ST and adverse pregnancy outcomes; which outcomes and how will they be assessed?
Thankyou for the question. Sample size was developed by a university biostatistician. As there was no epidemiologically studies around pituri use or its effects from which to estimate sample size, the statistician used the anecdotal evidence of observed oral quids in pregnant women presenting for birth (~30%) in this homogenous population with measurement of tobacco, nicotine and their metabolites as a primary outcome measure. The tobacco, nicotine and their metabolite data was obtained from research around smoked tobacco use in other pregnant populations. As we had no clear understanding of the effect of pituri on maternal and perinatal outcomes, all Table 2 data was collected and analysed as described in 3.4.
Do all births take place in hospital?
It is always desired that births occur in hospital. To facilitate that, remote dwelling women are transferred several weeks prior to birth to ‘sit down’ whilst awaiting the birth. The findings (not reported in this protocol), detail the in-hospital and out-of-hospital births.
Which outcomes has implications for the sample size calculations.
The primary measure was the tobacco, nicotine and their metabolite data and that was correlated to maternal and perinatal outcomes listed in Table 2 and the analysis is described in 3.4
Further, do all women in this geographic area attend the antenatal clinics? If not, can a description be provided regarding the differences between those who do attend and those who do not? It is likely that more at risk women may not attend the clinics (in remote locations, etc).
Thankyou for your interesting questions. Maternity services are provided in some communities directly by the on-site clinicians (midwives and medical officers) and by visiting clinicians (midwives, RFDS) or women are routinely taken to Alice Springs for antenatal care depending on their location. In the findings (not reported in this protocol paper), the number of antenatal visits is detailed.
Are there electronic records for births that do not take place in medical facilities?
Yes, all known births are recorded on CARESYS.
What were the compliance rates for the biological samples, and was there any bias introduced by women who agreed vs those who did not?
Thankyou for the question, the details of the number of samples collected in provided in the findings (not reported in this study protocol).
When was breast milk obtained? Usually milk does not fully come in until approximately 5-7 days post birth, so collection would need to be done in the home.
Thankyou for the question, the details of the number of samples collected in provided in the findings (not reported in this study protocol).
It is likely that the control variables (i.e. confounders) will differ by outcome. Please specifiy which variables for which analysis using a DAG.
Thankyou for the question that we spent much time on at the start of the study. We used the literature on known maternal and perinatal outcomes associated with maternal tobacco exposure to identify the first of the predictor, covariates and outcomes variables. However, given the unknown effects of pituri use and the uniqueness of the population (where endemic rates of adverse maternal and perinatal outcome predictors exist – some of which may be due to pituri use) we considered that there would be expected and unexpected interactions between tobacco use and tobacco and nicotine metabolite concentration and maternal and perinatal outcomes and the population demographics. We couldn’t be confident that a variable was only a predictor or only a confounder or only an outcome. Below is the extract from 3.4 that describes the variables and that were used to investigate predictor and/or covariate and/or outcome variables: For each biological sample, the nicotine and its metabolites cotinine, nicotine glucuronide, nicotine-N-oxide, nornicotine and trans-3’-hydroxycotinine were combined to provide a measure of nicotine exposure. Concentrations of anabasine and anatabine were used for comparison with self-reported tobacco use to indicate participant exposure (smoked or chewed) to tobacco plant material.
The association between tobacco use and maternal and neonatal outcomes was investigated and reported using the STROBE guidelines. Missing data and outliers are reported but are excluded from analysis. Outliers are identified by the use of boxplots and are excluded from data analysis based on established procedures. Results are reported with the level of significance at alpha .05 and accompanied by 95% confidence intervals (95% CI).
Descriptive and comparative analyses was undertaken using self-reported tobacco use as the independent variable. Means ± standard deviations or proportions are reported for in-group dispersion and central tendency where applicable. Continuous variables are tested for statistically significant differences between exposure groups using t-test and ANOVA, followed by appropriate post-hoc tests when required. Categorical variables are analysed by chi-square with appropriate post-hoc tests when required.
Multivariate logistic and linear regression, with adjustment based on previous literature examining tobacco use and pregnancy outcomes was conducted. Variables were measured as continuous or categorical (Table 4) and fitted as covariates in that manner in the regression models. Variables with a p < .2 in the regression model were accepted for covariate interaction inclusion.
a) Demographic variables (age, residential address, education level)
b) Lifestyle factors (alcohol and tobacco use)
c) Past and current medical history (cardiac, hypertension, diabetes and renal disease)
d) Pregnancy related factors (parity, gravida, elevated glucose, hypertension, STI, UTI, anaemia, number of antenatal visits, placental weight and size)
e) Labour and birthing factors (LUSCS, meconium staining, post-partum haemorrhage)
f) Birth outcome variables (gestational length, birthweight, gender, admission to SCN and APGAR score)
There were in fact completely unexpected interactions which are detailed in the findings (not reported in this study protocol).

Reviewer 3 Report
There does not seem to be anything (quasi) experimental about this study and neither is it longitudinal. It seems to be a prospective cohort study (observational). Additionally, the sampling technique as described is a convenient sample.
I would recommend including some comments on bias introduced as a result of choosing not to approach some women for inclusion in this study. This type of selection on the part of the researcher is concerning. Additionally, bias that may result as a result of informing the participants of the purpose of the study this late into the pregnancy. Some complications may already be evident and the responses of the women may be biased as a result.
For sample size calculation, to what does an effect size of 0.2 correspond in this study?
The Data analysis section is brief and lacks detail on rationale for the inclusion of covariates. Demographic factors such as age may confound the relationship between tobacco use and pregnancy outcomes. However, tobacco use may contribute to past and current medical conditions and thus be on the causal pathway.
In the Discussion the relevance of the phrase "trial design" is not clear since this is not a trial. Please clarify.
Author Response
Response to Review 2 comments (in black, italics font).
There does not seem to be anything (quasi) experimental about this study and neither is it longitudinal. It seems to be a prospective cohort study (observational). Additionally, the sampling technique as described is a convenient sample.
Thankyou for the comment, we used quasi in the sense that the participants were not randomly assigned to the groups. We have removed the words quasi experimental and longitudinal. We believe that the sampling approach did fit the pragmatic approach described by Hardon et al, and especially as we were mindful of the cultural nuances around approaching a potential participant in the presence of other people, however we will change the word to convenience sampling if the reviewer believes that it is a better description of the approach.
I would recommend including some comments on bias introduced as a result of choosing not to approach some women for inclusion in this study.
Thankyou, we have added your comment to the limitation of the study.
This type of selection on the part of the researcher is concerning.
The approach to participants was informed by a level of cultural understanding of the deep sensitivities and shame that can be generated when non-Aboriginal people thoughtlessly approach Aboriginal people. The approach utilises the guidelines from the NHMRC requirements for research with Aboriginal and Torres Strait Islander populations. While this recruitment strategy severely limited the number of potential approaches to approximately two-three per month, the approach did not favour or bias any tobacco-use group.
Additionally, bias that may result as a result of informing the participants of the purpose of the study this late into the pregnancy. Some complications may already be evident and the responses of the women may be biased as a result.
Thankyou for the comment. As active participants in this study, it was important for the women to know what the purpose of the study was and yes, there may have been bias introduced because of knowing what the purpose of the study was investigating and yes, complications may already have been evident at the time of enrolment. This would be applicable if the study enrolled women from 12 weeks or 20 weeks. The researchers considered there was less bias by enrolment after 28 weeks.
The participant questionnaire was related to social situation, their use of tobacco and alcohol and their knowledge of pituri preparation. No questions were related to the pregnancy.
For sample size calculation, to what does an effect size of 0.2 correspond in this study?
Thankyou for the comment. The measurement of tobacco, nicotine and their metabolites was a research question and compared non-tobacco use to cigarette use to pituri use. The effect size was related to that measurement and used by the biostatistician for the sample size calculation. That point has been clarified in the paper.
The Data analysis section is brief and lacks detail on rationale for the inclusion of covariates. Demographic factors such as age may confound the relationship between tobacco use and pregnancy outcomes. However, tobacco use may contribute to past and current medical conditions and thus be on the causal pathway.
Thankyou for the question. We completely agree with issue of what is a predicator, a covariate and an outcome. We spent much time on this matter at the start of the study. We used the literature on known maternal and perinatal outcomes associated with maternal tobacco exposure to identify the first of the predictor, covariates and outcomes variables (see first paragraph in 3.1) and Table 2.
However, given the unknown effects of pituri use and the uniqueness of the population (where endemic rates of adverse maternal and perinatal outcome predictors exist – some of which may be due to pituri use) we considered that there would be expected and unexpected interactions between tobacco use and tobacco and nicotine metabolite concentration and maternal and perinatal outcomes and the population demographics. We couldn’t be confident that a variable was only a predictor or only a confounder or only an outcome, thus the inclusion of a range of other variables including distance from health care, language, housing situation, income, educational level and significant adverse medical and obstetric history (Table 2).
Below is the extract from 3.4 that describes the variables and that were used to investigate predictor and/or covariate and/or outcome variables: For each biological sample, the nicotine and its metabolites cotinine, nicotine glucuronide, nicotine-N-oxide, nornicotine and trans-3’-hydroxycotinine were combined to provide a measure of nicotine exposure. Concentrations of anabasine and anatabine were used for comparison with self-reported tobacco use to indicate participant exposure (smoked or chewed) to tobacco plant material.
The association between tobacco use and maternal and neonatal outcomes was investigated and reported using the STROBE guidelines. Missing data and outliers are reported but are excluded from analysis. Outliers are identified by the use of boxplots and are excluded from data analysis based on established procedures. Results are reported with the level of significance at alpha .05 and accompanied by 95% confidence intervals (95% CI).
Descriptive and comparative analyses was undertaken using self-reported tobacco use as the independent variable. Means ± standard deviations or proportions are reported for in-group dispersion and central tendency where applicable. Continuous variables are tested for statistically significant differences between exposure groups using t-test and ANOVA, followed by appropriate post-hoc tests when required. Categorical variables are analysed by chi-square with appropriate post-hoc tests when required.
Multivariate logistic and linear regression, with adjustment based on previous literature examining tobacco use and pregnancy outcomes was conducted. Variables were measured as continuous or categorical (Table 2) and fitted as covariates in that manner in the regression models. Variables with a p < .2 in the regression model were accepted for covariate interaction inclusion.
a) Demographic variables (age, residential address, education level)
b) Lifestyle factors (alcohol and tobacco use)
c) Past and current medical history (cardiac, hypertension, diabetes and renal disease)
d) Pregnancy related factors (parity, gravida, elevated glucose, hypertension, STI, UTI, anaemia, number of antenatal visits, placental weight and size)
e) Labour and birthing factors (LUSCS, meconium staining, post-partum haemorrhage)
f) Birth outcome variables (gestational length, birthweight, gender, admission to SCN and APGAR score)
In the Discussion the relevance of the phrase "trial design" is not clear since this is not a trial. Please clarify.
Thankyou for the comment. Trial has been amended to research.

Round 2
Reviewer 2 Report
While the authors have addressed many of the prior comments, there are still a few that are concerning.
Sample size. There is adequate previous data to estimate an effect size (ie. reduction in birth weight) due to smoking cigarettes. I fully understand that there are no data on pituri use, however, one can use the smoking data to estimate the sample size.
Collection of breast milk. The question was not about how many samples will be collected but HOW they will be collected, as the authors want breast milk (not colostrum) which does not fully come in for 3-7 days after the birth. As this is a protocol paper, it is not unreasonable to ask for the method of collection of breast milk.
Covariate selection. It seems as though the authors have reported results elsewhere. Please give the reference. The important missing covariate (especially for birth weight) is prepregnancy weight of the mother (or height, which could be considered a proxy).
Author Response
Response to Reviewer 2 further remarks (black font)
While the authors have addressed many of the prior comments, there are still a few that are concerning. Sample size. There is adequate previous data to estimate an effect size (ie. reduction in birth weight) due to smoking cigarettes. I fully understand that there are no data on pituri use, however, one can use the smoking data to estimate the sample size.
Thank you for the question. This first study on pituri use in pregnancy had two aims, the first was the proof of principle i.e. the measurement of tobacco, nicotine and their metabolites in the participant population. We wanted to determine if there was evidence of clinically significant tobacco and nicotine exposure obtained from the maternal use of pituri and then to compare that to maternal use of cigarettes and non-tobacco exposure. There is no literature around nicotine metabolism pharmacokinetics and dynamics in central Australian Aboriginal populations, however we do know that there are differences in nicotine metabolism based on genetic profile, and we do know that in pregnancy nicotine is metabolized differently. In determining the population sample, we had a proportional guesstimate of about 0.3 for pituri use and the same for smoking (from the NT perinatal data records); the statistician considered the populations’ reduced epigenetic variability in terms of nicotine metabolism and the sensitivity and specificity of the testing mechanisms and the age of the population (18-50). Using ANOVA for three groups, with an expected large difference between the control and the two tobacco groups on tobacco and nicotine and their metabolites, a power of 80% and alpha 0.05 provided a sample population of 54. The homogeneity of the population further reduced this sample size. In the research, data was collected from 74 participants to allow for missing data elements.
The second aim of the study was the maternal and neonatal outcomes correlated to the maternal tobacco exposure. We agree completely with the reviewer that birthweight is one variable that is used in comparison of the neonates of smoking mothers to non-smoking mothers, however we were conscious that other maternal variables (for example the presence of elevated glucose and hypertension – which often occur at higher rates in Aboriginal populations) and gestational age also impact birthweight.
The outcomes from the study are being prepared for publication (following the publication of this protocol) and show that if the study had focused on maternal and perinatal outcomes (for example birthweight and gestational age), significant associations with tobacco and nicotine concentrations and pregnancy outcomes would not have become apparent.
Collection of breast milk. The question was not about how many samples will be collected but HOW they will be collected, as the authors want breast milk (not colostrum) which does not fully come in for 3-7 days after the birth. As this is a protocol paper, it is not unreasonable to ask for the method of collection of breast milk.
Thank you, we appreciate reviewers comment and have changed the wording to ‘colostrum and/or breast milk’. Of note, it is the normal practice for remote dwelling women stay in hospital for longer periods than town dwelling women post-birth.
Covariate selection. It seems as though the authors have reported results elsewhere. Please give the reference. The important missing covariate (especially for birth weight) is pre-pregnancy weight of the mother (or height, which could be considered a proxy).
The authors completely agree with the reviewer that pre-pregnancy weight and BMI are important variables and Table 2 lists BMI as a variable to be collected in the research.
The detailed maternal and neonatal findings will be provided in subsequent publication/s and are not reported in this study protocol. However other aspects of the research have been published (please see links) below.
https://www.academia.edu/19710450/The_pituri_story_a_review_of_the_historical_literature_surrounding_traditional_Australian_Aboriginal_use_of_nicotine_in_Central_Australia
https://www.sciencedirect.com/science/article/pii/S240584401732073X?via%3Dihub
https://link.springer.com/article/10.1007/s00038-014-0558-6
https://www.rrh.org.au/journal/article/4044